# Determining How Far an Adult Rare Disease Patient Needs to Travel for a Definitive Diagnosis: A Cross-Sectional Examination of the 2018 National Rare Disease Survey in China

**DOI:** 10.3390/ijerph17051757

**Published:** 2020-03-08

**Authors:** Xiang Yan, Shenjing He, Dong Dong

**Affiliations:** 1Department of Urban Planning and Design, Faculty of Architecture, The University of Hong Kong, Pokfulam, Hong Kong, China; shawnyan@connect.hku.hk (X.Y.); sjhe@hku.hk (S.H.); 2Shenzhen Institute of Research and Innovation, The University of Hong Kong, Shenzhen 518057, China; 3JC School of Public Health and Primary Care, Faculty of Medicine, The Chinese University of Hong Kong, Hong Kong, China

**Keywords:** rare disease, adult, diagnosis, accessibility, healthcare, China

## Abstract

**Background:** To investigate the multidimensional difficulties in accessing a definitive diagnosis of adult rare diseases and the associated impact factors in China. **Methods:** A total of 1010 adult rare disease patients from the 2018 China Rare Disease Survey were used for analysis. The Structural Equation Models examined the interrelationships among five accessibility indicators and the effects of three sets of impact factors. **Results:** (1) Accessibility: 72.97% of patients were misdiagnosed; they waited an average of 4.30 years and visited 2.97 hospitals before the definitive diagnosis; 67.13% were diagnosed outside the home city and traveled an average of 562 km. (2) Interrelationships among accessibility indicators: the experience of misdiagnosis significantly increased diagnosis delay and the number of hospitals visited, but had no significant effect on healthcare utilization across cities. (3) Impact factors: the rarity of disease only increased the number of hospitals visited and residence–hospital distance; high-quality healthcare distribution was key in determining accessibility; the older, disabled, poor, and less-educated individuals, and those in Central/West China were disadvantaged. **Conclusion:** The socioeconomic dimension of difficulties in accessing a definitive diagnosis of rare diseases should be attended, especially the uneven distribution of high-quality healthcare and those disadvantaged patients. More systematic rare disease surveys are needed in the future.

## 1. Introduction

Rare diseases are not that rare. Although every single rare disease affects only an extremely limited number of patients (defined as incidence <1/2000 in Europe and <1/1250 in the United States), approximately 6000–8000 disorders are classified as rare diseases along with 250–280 new additional ones annually, which accounts for 6%–10% of the global population being affected [1,2]. Among the numerous and varied problems experienced by rare disease patients and their families, the first and perhaps most significant problem that prevents them from achieving a better quality of life is the difficulty in accessing a definitive diagnosis. A definitive diagnosis does not only mean a possible treatment and relief from pain, but also means release from pressure of not knowing, access to ancillary social welfare or subsidies for special needs, connection with rare disease support groups, and obtaining information for life planning and reproductive decision-making [3,4]. Misdiagnosis, diagnosis delay, and a lengthy journey to diagnosis are commonly experienced by rare disease patients. A 2006 study of eight rare illnesses in 17 European nations revealed that up to 25% of patients had spent 5–30 years to access the correct diagnosis, and 40% of the patients experienced an erroneous diagnosis [5]. Recognizing the essential but undoubtedly challenging task on improving accessibility to rare disease diagnosis, the International Rare Diseases Research Consortium has set out its vision to enable all people living with a rare disease to receive a definitive diagnosis within one year of coming to medical attention [6]. 

Medical endeavors are essential to achieve this goal, such as improvements in gene techniques [7]. However, the impact of the geographic distribution of healthcare and patient mobility should also be taken into consideration. Due to the uneven distribution of healthcare, many patients have to travel across regions to obtain a definitive diagnosis. Eurordis’ survey revealed that 25% of rare disease patients in Europe had to travel to a different region to receive the definitive diagnosis, and 2% even had to travel to a different country [8]. The variance of accessibility to healthcare can also be caused by differences in patient mobility, such as affordability, physical disability, and education level [8,9,10,11]. In general, people with lower socioeconomic status may find it more difficult to access healthcare. For patients with severe disease like cancers, lack of knowledge and patients’ disability may lead to a worse prognosis [12,13]. However, in the case of rare disorders where the diagnosis may be largely beyond the patient’s efforts, it remains unclear to what extent the patient-related factors affect.

In China, approximately 20 million people are afflicted with rare diseases [14]. Efforts to cope with rare diseases can be traced back to the 1980s, with substantial advances in epidemiology, case registration, basic research, the establishment of medical networks, and orphan drugs during these years [15]. However, much work is needed in response to the accelerating awareness of rare disorders in China. The first national survey on rare diseases in 2016, covering 1771 patients across the country, reported that the various social and economic difficulties faced by these people are “beyond imagination” [16]. Traveling far for a definitive diagnosis is a common scenario in China, not only because of the limited medical understanding of rare disorders but also the uneven distribution of high-quality healthcare. In China, high-quality healthcare concentrates in 3-A hospitals, which are unevenly distributed across the country [17]; hence, many patients have to travel across cities to find a definitive diagnosis. Those people who have to depend on trans-urban diagnosis may experience a tougher journey.

Although the difficulties of accessing a definitive diagnosis for rare diseases in China are widely perceived and acknowledged, there has been no systematic study yet. A systematic and quantitative analysis is essential for policymaking and improving social support to rare disease patients. Hence, this study aimed to investigate the multidimensional difficulties in accessing a definitive diagnosis of adult rare diseases and the associated impact factors in China.

## 2. Materials and Methods

### 2.1. Survey on Rare Disease Patients

The 2018 China Rare Disease Survey investigated the patient’s accessibility to a definitive diagnosis at a national level. For each patient, data on the disease, demographic characteristics, health condition, and socioeconomic features were collected. Questions about patient’s journey to the definitive diagnosis were asked, including experience of having been misdiagnosed, the time when they started seeking a diagnosis and the time they received a definitive diagnosis, the number of hospitals visited prior to receiving a definitive diagnosis, and name of the hospital where their diagnosis was confirmed.

The online questionnaires reached the widely spread patients across the country with the assistance of the Illness Challenge Foundation, a national rare disease umbrella patient organization working together with multiple patient groups focusing on a single rare disease. Finally, 2040 valid questionnaires were collected in China, and data of 1135 adult patients (defined as those aged at least 18 years in 2018) were collected. Considering the differences between adult and child patients in terms of incidence of diseases and patients’ ability to act on their own, this study focused on adult patients only. A total of 1010 individuals in Mainland China with complete information on each item were included in the analysis, accounting for 89.1% of the total samples. These 1010 individuals represented a large community of patients with 63 rare diseases. In descending order, the 10 diseases with the most number of reported cases are Behcet’s disease, Acromegaly, Fabry disease (FD), Peutz-Jeghers syndrome (PJS), Glycogen storage disease, Pseudomyxoma peritonei, Multiple sclerosis, Pulmonary hypertension, Neuromyelitis optica, and Osteogenesis imperfecta, accounting for 66.8% of the total cases.

Figure 1 is a heat map displaying the distribution of the cases. The map was generated using software package ArcGIS 10.3 (so were Figures 2 and 4). A large proportion of cases are distributed in East China, particularly in the Beijing-Tianjin-Hebei, Yangtze River Delta, and Pearl River Delta Urban agglomerations. The distribution of cases in Central and West China are highly dispersed, except in some provinces like Sichuan and Hubei. This spatial pattern is a result of both the distribution of patients and the networks of rare disease patient groups through which samples were collected.

### 2.2. Measurement of Accessibility to Definitive Diagnosis

Diagnosis delay is commonly used to measure the accessibility to diagnosis. However, as studies in the EU and the U.S. stressed [5,8,18], accessing the diagnosis of rare diseases is much more complicated. Besides diagnosis delay, rare disease patients also frequently experienced misdiagnosis, had to consult many hospitals, and traveled a long distance, usually across cities and sometimes across countries. Rare disease patients in China faced similar difficulties. Resonating with studies in the EU and the U.S., we therefore used five indicators of accessibility in this study to measure the accessibility to a definitive diagnosis in a fuller manner: (1) experience of having been misdiagnosed; (2) diagnosis delay, defined as the time elapsed from the first visit to healthcare until definitive diagnosis was obtained; (3) number of hospitals visited prior to obtaining the definitive diagnosis; (4) whether the diagnosis was trans-urban, which is meaningful as many patients have to travel across cities because of the uneven distribution of healthcare; and (5) residence–hospital distance, measured by the distance between home city and the city where patients obtained the definitive diagnosis (the distance was 0 when patient received the definitive diagnosis in home city).

### 2.3. Impact Factors of Accessibility

Andersen’s healthcare utilization model has been extensively used to study patient access to healthcare [19,20], especially the diagnosis delay and patient choice of healthcare. According to the model, access to healthcare is determined by healthcare system (policy, resources, and organization) and population characteristics (predisposing factors, enabling factors, and need). 

With regard to healthcare system, we mainly considered the impact of geographical distribution of high-quality healthcare resources in China, because it is a dominant healthcare factor determining patient access to a definitive diagnosis. Factors on the policies and organizations on rare diseases was not included in this study because China is still catching up on those issues compared to regions like the EU and the U.S. In China, healthcare services are provided in primary care institutes, public health institutes, and hospitals. Different from the referral systems in many other countries, patients in mainland China can directly seek healthcare in any hospitals on their own, which may or may not be covered by their medical insurance. The hospitals are further classified into three classes, with the 3-A hospitals at the highest level with the best equipped facilities and most adequately trained medical staff [21]. Since the diagnosis of rare diseases often requires higher levels of experience and more advanced diagnostic technologies, most patients resort to 3-A hospitals for diagnosis. We used the number of 3-A hospitals in each city to measure the geographic distribution of high-quality healthcare. We hypothesize that the more 3-A hospitals in a patient’s home city, the better the accessibility to a definitive diagnosis. Most studies evaluating China’s healthcare distribution rely on the statistics provided by the National Health and Family Planning Commission (NHFPC, renamed as National Health Commission after 2018), neglecting the fact that not all healthcare resources are under its administration. Hospitals managed by the Ministry of Public Security, People’s Liberation Army, and other institutions account for a considerable percentage of high-quality healthcare [17]. We built a fuller hospital database to include as many 3-A hospitals as possible by combing different data sources, including statistical yearbooks, government websites, hospitals’ official websites, and Internet medical platforms (such as www.haodfu.com). Data from different data sources have been cross-validated. Finally, the database covered 1295 3-A hospitals in Mainland China. Although there are some possible omissions, the number is much bigger than the 705 3-A hospitals in the list announced by NHFPC in 2017. Figure 2 shows the uneven distribution of 3-A hospitals on the city level. The cities with most 3-A hospitals are Beijing (60) and Shanghai (50). 

With regard to the impact of patient characteristics, we extracted several key factors from the commonly studied demographic and socioeconomic status characteristics to assess patient mobility, because most rare disease patients have to travel across hospitals and places for diagnosis. Five groups of factors of patient mobility are considered in this study: (1) family income level, to measure patient affordability to healthcare; (2) patient physical disability, measured by the degree of dependency on assistive devices; (3) education level, which affects patient ability to access data of target healthcare; (4) sex and age; and (5) patient residential region. According to studies in Europe, higher affordability, less physical disability, and more education are associated with better patient mobility [8,9,10,11]. Patient residential region was considered because East China has much better transportation and a higher concentration of high-quality healthcare than Central and West China, which may shorten the diagnosis delay and traveling distance to the definitive diagnosis.

In the case of rare disease, rarity of disease was used as an additional impact factor. The classic frameworks on accessibility only emphasize the influences of healthcare system and patient characteristics [19,20,22,23]. However, studies on other severe diseases like cancers have suggested that the type of disease should be embraced as an irrespective factor affecting access to healthcare [24]. Although the prevalence of all rare diseases is low, the degree of prevalence still varies significantly. We constructed the variable “rarity of disease” by categorizing the diseases into three classes based on the reported prevalence of each disease: “extremely rare” with an incidence below 1/100,000, “rare” with an incidence ranging from 1/100,000 to 1/10,000, and “somewhat rare” with an incidence above 1/10,000. The prevalence data were mainly obtained from Orphanet (orpha.net), an international knowledge platform for rare diseases. Data on the prevalence of 20% of diseases were obtained from published academic papers as their information was not available in Orphanet. The details of prevalence and data sources are listed in Table A1.

### 2.4. Conceptual Framework of Structural Equation Model

The five indicators of accessibility to a definitive diagnosis are interrelated. Based on our observation, we built a framework to conceptualize their interrelationship, as shown in the left box of Figure 3. We hypothesized that patients usually first seek a diagnosis in local hospitals. If they cannot be accurately diagnosed locally, they have to spend more time and travel to more hospitals. As time passes by and the number of local hospitals they visit increases, they may seek diagnosis outside of the home city and thus travel further. The reason to use Structural Equation Model is that it allows the investigation of multiple interrelationships among accessibility indicators and the direct/indirect effects of impact factors on accessibility. All nominal variables, such as age and misdiagnosis, are used as ordinal variables in the SEM model. The model was run in AMOS, version 24.0. The model was estimated using the maximum likelihood method, and the goodness-of-fit was evaluated based on the widely used criteria [25]: GFI, TLI, CFI > 0.9, RMESA < 0.06, SRMR < 0.08, and chi-square/degree of freedom (*x*^2^/*df*) < 3.0.

### 2.5. Ethical Approval

This study was approved by the Committee on the Use of Human and Animal Subjects in Teaching and Research of Hong Kong Baptist University, which the corresponding author worked for when the study was conducted. The ethics approval code is FRG2/15-16/052.

## 3. Results

### 3.1. Description Analysis

Table 1 illustrates the accessibility to a definitive diagnosis for adult rare disease patients. The proportion of patients who were misdiagnosed was 72.97%. The average diagnosis delay was 4.3 years. Approximately 48.10% of patients obtained diagnosis within the first year, while 14.30% spent more than 10 years, suggesting the significant disparities among rare disease patient groups. The number of hospitals visited prior to diagnosis was 2.97; approximately 23.3% of the patients obtained a diagnosis from the first hospital they visited, and 4.4% visited more than 10 hospitals. 

Only 32.87% of patients were accurately diagnosed in the cities where they live. Generally, cities with more 3-A hospitals were associated with a higher proportion of local diagnosis. For example, 86.15% of patients in Beijing and 82.86% of patients in Shanghai received definitive diagnoses in local hospitals. For the remaining 67.13% of patients, who had to travel across cities and more often provinces to seek a definitive diagnosis, the average distance between house address and the hospital for definitive diagnosis was 562.72 km. Approximately 12.08% of the 1010 patients traveled over 1000 km (the distance between Beijing and Shanghai). Figure 4 maps the paths of patients traveling across cities to seek a definitive diagnosis. The top destinations of trans-urban diagnoses were Beijing, Shanghai, and Guangzhou, where high-quality medical services are offered and whose hospitals delivered definitive diagnosis for 31.19%, 11.98%, and 5.54% of patients, respectively. Other cities, especially provincial capitals such as Jinan, Wuhan, Chengdu, Hangzhou, Xi’an, Changsha, Chongqing, Nanjing, Tianjin, also played an important role.

Table 2 shows the description of impact factors. Among the 1010 adult rare disease patients, 46.63% were men, and the average age was 35.51 years. Approximately 44.36% of them were physically disabled and had to depend on assistive devices. Moreover, 63.76% of families suffered an income below the mid-level of the local city (based on self-assessment by patients or their caregivers). The average education level was 12.05 years (equivalent to senior high school education). Approximately 49.50% of families were from East China. The average number of 3-A hospitals in patients’ cities was 12.60. There were 8.51% of patients afflicted with extremely rare diseases, and 15.75% afflicted with somewhat rare diseases.

### 3.2. Results of SEM Analysis

The model was a good fit to the data on the criteria (*x*^2^/*df* = 2.196, GFI = 0.993, TLI = 0.920, CFI = 0.978, RMESA = 0.034, and SRMR = 0.024). The model accounted for 1.4%, 4.8%, 8.4%, 27.0%, and 34.2% of the variance in experience of having been misdiagnosed, diagnosis delay, the number of hospitals visited, whether the diagnosis was trans-urban, and residence–hospital distance, respectively. The direct, indirect, and total effects are presented in Table 3.

Firstly, we analyzed the interrelationships between accessibility indicators to determine the healthcare utilization pattern of adult rare disease patients (Figure 5). The experience of having been misdiagnosed had a significant effect on both diagnosis delay and the number of hospitals visited, with a standardized effect of 0.189 and 0.244, respectively. If the patients were misdiagnosed, they spent 2.44 more years and visited 2.59 more hospitals. In contrast to our hypothesis, the experience of having been misdiagnosed did not have a direct effect on trans-urban diagnosis. Among the 273 patients who did not experience being misdiagnosed, 60.44% were accurately diagnosed outside their cities. However, the experience of having been misdiagnosed had an indirect impact on trans-urban diagnosis mediated by the number of hospitals visited, with a standardized effect of 0.023. The trans-urban diagnosis had a significant impact on residence–hospital distance, with a high standardized effect of 0.569. The residence–hospital distance was also indirectly affected by the experience of having been misdiagnosed and the number of hospitals visited, with a standardized effect of 0.039 and 0.043, respectively. Surprisingly, diagnosis delay had no significant effects on trans-urban diagnosis and residence–hospital distance. Take the relationship between diagnosis delay and trans-urban diagnosis as an example. Among the 486 patients who obtained a definitive diagnosis within the first year, 62.75% were diagnosed outside their city. On the contrary, among the 45 patients who waited for four years to obtain a definitive diagnosis, only 55.56% were diagnosed outside their city.

Secondly, we examined the effect of impact factors on accessibility indicators (Figure 6). The rarity of disease only had a significant impact on the number of hospitals visited and the residence–hospital distance. If the disease was rarer, the patients had to visit more hospitals and traveled further. However, no significance was found in the effects of the rarity of disease on the other three accessibility indicators. Less rare diseases were also associated with a higher proportion of misdiagnosis and longer diagnosis delay. Take the comparison between PJS and FD, for example. PJS is an extremely rare disease with a prevalence of 1–9/1,000,000, and FD is about 1/100,000 (see Appendix A for the details). For PJS (68 cases), the proportion of patients who were misdiagnosed was 73.53%, and the average diagnosis delay was 6.84 years. For FD (68 cases), the proportion of patients who were misdiagnosed was higher at 94.12%, and the average diagnosis delay was 9.68 years. The rarity of disease was also not significantly related to trans-urban diagnosis.

The distribution of 3-A hospitals had significant impacts on the number of hospitals visited, trans-urban diagnosis, and residence–hospital distance. The more 3-A hospitals in a patient’s city, the more hospitals the patient would visit but with less necessity of trans-urban diagnosis and shorter traveling distance. In particular, the absolute value of its standardized effect on trans-urban diagnosis was as large as 0.494, suggesting that the distribution of 3-A hospitals determined nearly half of trans-urban diagnosis. Nevertheless, no significant effects were observed on the experience of having been misdiagnosed and diagnosis delay.

With regard to patient mobility, each factor only had significant effects on one or two accessibility indicators. Female patients traveled a shorter distance compared with male patients, although the absolute value of standardized effect was low at 0.07. Older patients were more likely to obtain a definitive diagnosis in the home city. If the degree of disability was higher, they were more constrained to be diagnosed in the home city and thus traveled a shorter distance. Family income level was the only factor that significantly influenced the experience of having been misdiagnosed. The richer the family, the more likely they were able to access better healthcare in the first place and the less likely the possibility of being misdiagnosed. It also had negative significant indirect impacts on diagnosis delay and the number of hospitals visited.

Contrary to our hypothesis, the education level had no significant effect on the experience of having been misdiagnosed and diagnosis delay. Put differently, patients with higher education levels did not perform better in seeking the right healthcare than those with less education. However, the education level positively influenced the residence–hospital distance, suggesting patients with better education would travel further for a definitive diagnosis. Consistent with our hypothesis, the patient’s residential region had a significant impact on the residence–hospital distance; patients in East China traveled much less, and those in Central and West China were disadvantaged. The standardized effect was 0.218, which is higher than all the other five factors. According to the total effects, one more interesting finding is that only the diagnosis delay was not significantly affected by all of the six patient mobility factors.

## 4. Discussion

Based on the 2018 China Rare Disease Survey, this is the first study to evaluate the accessibility to a definitive diagnosis of adult rare disease patients in China. The significance of a definitive diagnosis not only lies in possible treatment, but also in enabling patients and their caregivers to enter into various public welfare and social support networks. In this study, we not only investigated the multidimensional difficulties adult rare disease patients experienced when seeking diagnosis, but also used the Structural Equation Model to examine the healthcare utilization pattern and the associated impact factors. This study contributes to the rising concern on the livelihood of rare disease patients.

The survey revealed the great but uneven difficulties faced by adult rare disease patients in Mainland China. Approximately 72.97% of patients received an erroneous diagnosis, which is much higher than the result of Eurodis’ 2006 survey on eight rare diseases, five of which were included in our study (Crohn’s disease, Duchenne muscular dystrophy, Marfan syndrome, Prader-Willi syndrome, and tuberous sclerosis), while three were not (Cystic fibrosis, Ehlers-Danlos syndrome, and Fragile X syndrome) [5,8]. This contrast implies that China faces a tougher situation in coping with rare disorders. On average, patients had to wait 4.30 years and visit 2.97 hospitals prior to the definitive diagnosis. However, the difficulties varied among patients. Approximately 48.10% of patients were accurately diagnosed within the first year and 23.3% in the first hospital, while 14.3% had to wait more than 10 years and 8.8% had to visit more than 10 hospitals. Similar unevenness was also found in Europe [8], calling for more attention to the equity issue in the diagnosis of rare disease. This unevenness also existed in the geographical dimension. Approximately 67.13% of patients had to travel across cities for a definitive diagnosis, with an average traveling distance of 562.72 km. Nearly half of patients who failed to be accurately diagnosed in their cities traveled to Beijing, Shanghai, and Guangzhou.

The SEM analysis increased the understanding of the healthcare utilization pattern of adult rare disease patients, such as whether having been misdiagnosed was a central issue, with a significant direct or indirect effect on all other four accessibility indicators. Patients who experienced misdiagnosis would visit more hospitals and travel across cities for diagnosis. However, misdiagnosis had no significant direct impact on trans-urban diagnosis, suggesting that not all patients go to other cities only after being misdiagnosed in local hospitals. Some may have access to high-quality healthcare outside their city in the first place. One more interesting finding is that diagnosis delay had no effect on trans-urban diagnosis and residence–hospital distance. Some patients may have access to healthcare outside their city at the very early stage, while some would not even try after waiting for many years.

Contrary to the hypothesis that the rarity of disease would affect all accessibility indicators, it only had a significant effect on the number of hospitals visited and residence–hospital distance, while not on the experience of having been misdiagnosed, diagnosis delay, and trans-urban diagnosis. This is strong evidence of the necessity to look into the socioeconomic difficulties faced by rare disease patients, although the medical endeavors are still primary and essential in coping with rare disorders.

The distribution of 3-A hospitals is a key element in determining the accessibility to a definitive diagnosis. Patients in a city with more 3-A hospitals would travel less but still could visit many hospitals for consultation. The uneven distribution of 3-A hospitals determined nearly half of trans-urban diagnosis. However, the experience of having been misdiagnosed and diagnosis delay were not significantly influenced by the distribution of 3-A hospitals. This finding highlights the underdeveloped medical network on rare diseases even in 3-A hospitals. According to China’s Rare Disease Report 2018 [26], skilled doctors had a lesser understanding of rare diseases. Among the 285 doctors who were interviewed (of whom 77% had a Master or Doctor degree and 76% had worked for more than five years), only 23.86% acknowledged having some or a good understanding of the rare disease, while 33.33% only heard about it but barely had any knowledge. For the future, both the medical network of rare disease and the unevenness of high-quality healthcare distribution should be attended.

More profound socioeconomic disparities were found in the influence of patient mobility. Sex, age, physical disability, family income, education level, and residential region all had significant effects on one or two accessibility indicators. Generally speaking, individuals who were considered as disadvantaged were the older, disabled, poor, and less educated patients, and those who lived in Central and West China. Among the six factors, family income level was the only one that had an influence on the experience of having been misdiagnosed, suggesting the rich patients’ advantage in accessing better healthcare in the first place. Contrary to our hypothesis, a higher education level could not help to reduce the possibility of being misdiagnosed. This may be because rare diseases were much beyond the patient’s knowledge. The diagnosis delay was not significantly affected by any of the patient mobility factors, which indicated that the diagnosis delay was much beyond patients’ endeavors even if they travel across cities and visit many hospitals.

This study has three main limitations. The first is the nonprobability sampling method coupled with a limited sample size may introduce the risk of sampling bias to our study. The second limitation is the presence of geographical bias. Although the 1010 cases have thoroughly covered all 31 provinces in Mainland China, there are only a small number of cases in some provinces such as Tibet, Qinghai, Yunnan, and Hainan. The difficulties in accessing a definitive diagnosis faced by patients in these provinces may not be well represented. The third limitation is bias in disease types. Although 63 diseases were included in this study, many rare diseases common in China were not included, and some diseases only had a very limited number of cases. However, as systematic investigations on rare disease patients are rare, this study can contribute some pioneering understanding. The findings could provide social and policy support to rare disease patients. Coping with the “not rare” rare disease faced by 20 million patients in China, there is an urgent need to have more systematic surveys and investigations on their pattern of accessing diagnosis and various difficulties they faced.

## 5. Conclusions

A definitive diagnosis is the first step to improve the livelihood of rare disease patients and their families. The difficulties in accessing a definitive diagnosis are multidimensional, and include misdiagnosis, long waiting time, and the need to visit many hospitals and travel across cities for diagnosis. The conclusions of this study are three-folded. (1) The 2018 China Rare Disease Survey is the first to investigate the accessibility to definitive diagnosis in China. The 1010 cases account for 72.97% adult rare disease patients who have been misdiagnosed. The average diagnosis delay was 4.30 years. The patients visited 2.97 hospitals before the definitive diagnosis. Approximately 67.13% of patients obtained their definitive diagnosis outside the home city, and these patients traveled an average distance of 562 km. (2) The healthcare utilization pattern of adult rare disease patients was explored. The experience of having been misdiagnosed significantly increased diagnosis delay and the number of hospitals visited. Nevertheless, healthcare utilization across cities was influenced by neither the experience of misdiagnosis nor the diagnosis delay. (3) The effect of impact factors on accessibility to a definitive diagnosis was examined. The rarity of disease only played a limited role in affecting accessibility to a definitive diagnosis. It only had significant effects on the number of hospitals visited and residence–hospital distance, while not on the experience of misdiagnosis, diagnosis delay, and trans-urban diagnosis. The distribution of 3-A hospitals was a key factor influencing accessibility. The uneven distribution of high-quality healthcare in China should be attended. The patients who were less likely to obtain a definitive diagnosis were the older, disabled, poor, and less educated individuals, and those who lived in Central and West China. To cope with rare disease, more attention should be paid to the livelihood of rare disease patients in addition to medical issues. Moreover, more systematic surveys are needed in the future to further understand the accessibility to the diagnosis of rare diseases in China.

## Figures and Tables

**Figure 1 ijerph-17-01757-f001:**
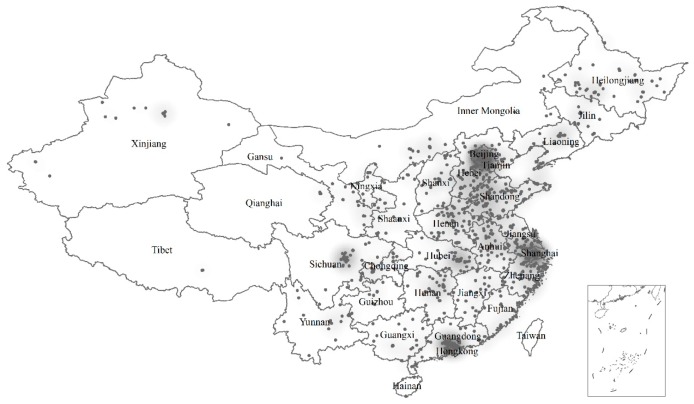
The distribution of 1010 adult rare disease patients in China.

**Figure 2 ijerph-17-01757-f002:**
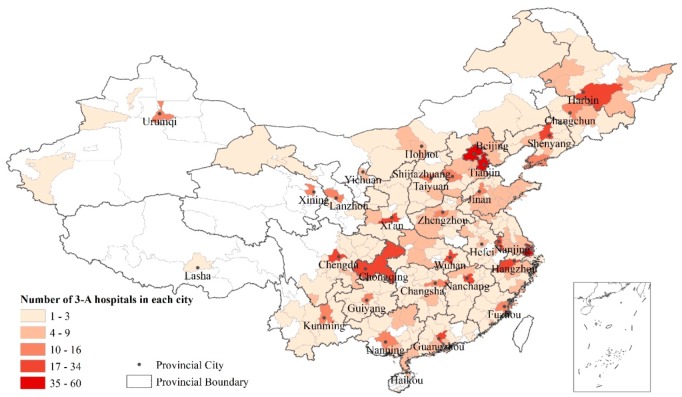
The distribution of 3-A hospitals in Mainland Chinese cities.

**Figure 3 ijerph-17-01757-f003:**
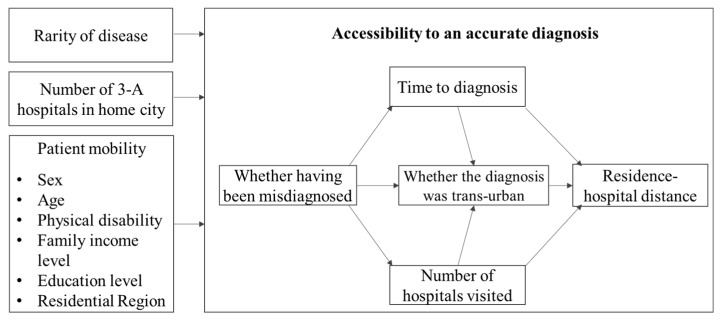
Conceptual framework of SEM analysis.

**Figure 4 ijerph-17-01757-f004:**
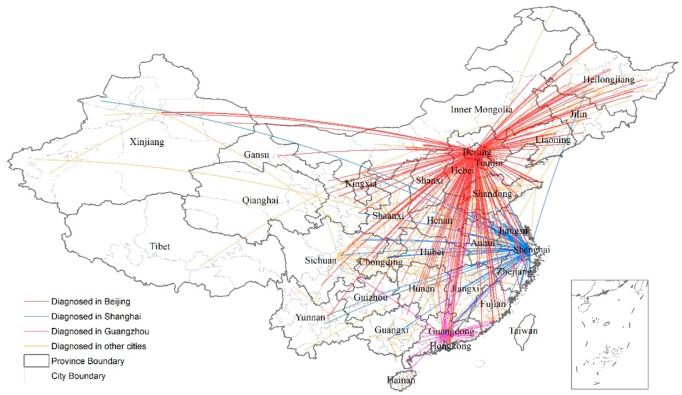
Traveling paths of patients seeking a definitive diagnosis in Mainland China.

**Figure 5 ijerph-17-01757-f005:**
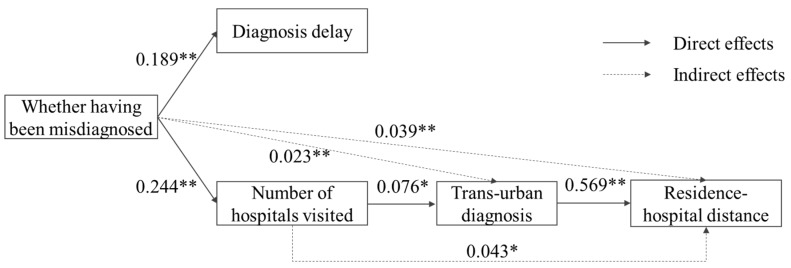
Direct and indirect effects between accessibility indicators with standardized coefficients. Note: * *p* < 0.05, ** *p* < 0.01.

**Figure 6 ijerph-17-01757-f006:**
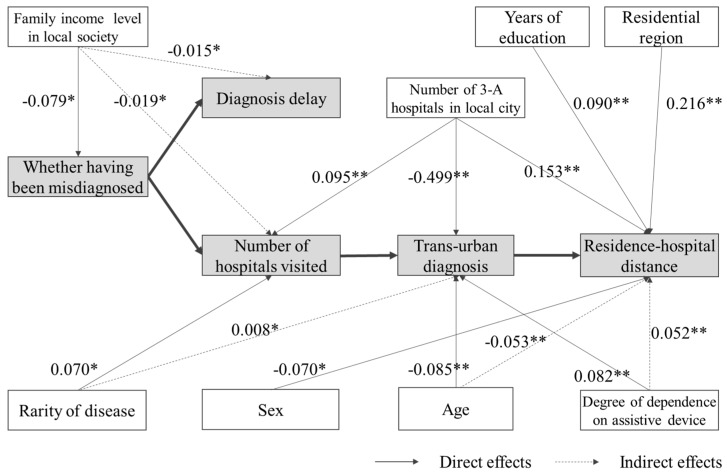
Direct and indirect effects of impact factors on accessibility. Note: * *p* < 0.05, ** *p* < 0.01. Effects of sex on Residence–hospital distance is the total effect.

**Table 1 ijerph-17-01757-t001:** Accessibility to diagnosis for rare disease patients in China.

Indicators	Mean	S.D.	Median	Range
Proportion of patients having been misdiagnosed	72.97%	-	-	-
Diagnosis delay (years)	4.30	5.75	2	1–44 ^1^
Number of hospitals visited prior diagnosis	2.97	4.71	2	1–80
Proportion of trans-urban diagnosis	67.13%	-	-	-
Residence–hospital distance (km)	374.41	502.91	170.51	0–3728.18
Residence–hospital distance of trans-urban diagnosis (km)	562.72	523.63	383.19	23.30–3728.18

Note: ^1^ The value 1 refers to patients who were accurately diagnosed in the first year of seeking the diagnosis.

**Table 2 ijerph-17-01757-t002:** Description of impact factors.

	Mean (S.D.)
**Sex**	
* Male*	46.63%
* Female*	53.47%
**Age**	35.51 (11.13)
**Frequency of using assistive device (Degree of physical disability)**	
* Always*	8.32%
* Usually*	7.62%
* Sometimes*	10.10%
* Occasionally*	18.32%
* No need*	55.64%
**Family income level in local city**	
* Low*	23.76%
* Relatively low*	40.00%
* Mid-level*	32.67%
* Relatively high*	3.37%
* High*	0.20%
**Years of education**	12.05 (4.12)
**Residential region**	
* East China*	49.50%
* Central China*	22.87%
* West China*	27.63%
**Number of 3-A hospitals in local city**	12.60 (16.76)
**Rarity of disease**	
* Extremely rare*	8.51%
* Rare*	75.74%
* Somewhat rare*	15.75%

**Table 3 ijerph-17-01757-t003:** Direct, indirect, and total effects of the model.

	Whether Having Been Misdiagnosed			Diagnosis Delay			Number of Hospitals Visited			Whether Diagnosis Was Trans-Urban			Residence–Hospital Distance		
	Direct	Indirect	Total	Direct	Indirect	Total	Direct	Indirect	Total	Direct	Indirect	Total	Direct	Indirect	Total
Whether having been misdiagnosed *(Not = 0)*	-	-	-	2.444 ** (0.189)	-	2.444 ** (0.189)	2.591 ** (0.244)	-	2.591 ** (0.244)	0.023 (0.022)	0.024 * (0.023)	0.047 (0.045)	-	44.564 * (0.039)	44.564 * (0.039)
Diagnosis delay	-	-	-	-	-	-	-	-	-	0.002 (0.024)	-	0.002 (0.024)	0.114 (0.001)	1.173 (0.013)	1.286 (0.015)
Number of hospitals visited	-	-	-	-	-	-	-	-	-	0.008 * (0.076)	-	0.008 * (0.076)	5.977 (0.056)	4.602 * (0.043)	10.579 * (0.099)
Whether diagnosis was trans-urban *(Diagnosed in local city = 0)*	-	-	-	-	-	-	-	-	-	-	-	-	610.639 ** (0.569)	-	610.639 ** (0.569)
Rarity of disease *(Somewhat rare = 0)*	0.017 (0.019)	-	0.017 (0.019)	0.752 (0.064)	0.042 (0.004)	0.794 (0.067)	0.680 * (0.070)	0.044 (0.005)	0.724 * (0.075)	0.032 (0.033)	0.007 * (0.008)	0.039 (0.041)	41.481 (0.040)	28.312 (0.028)	69.793 * (0.068)
Number of 3-A hospitals in local city	−0.002 (−0.062)	-	−0.002 (−0.062)	0.017 (0.050)	−0.004 (−0.012)	0.013 (0.038)	0.027 ** (0.095)	−0.004 (−0.015)	0.023 ** (0.080)	−0.014 ** (−0.499)	0.000 (0.006)	−0.014 ** (−0.494)	4.596 ** (0.153)	−8.288 ** (−0.276)	−3.692 ** (−0.123)
Sex *(Male = 0)*	−0.029 (−0.033)	-	−0.029 (−0.033)	−0.617 (−0.054)	−0.072 (−0.006)	−0.688 (−0.060)	−0.142 (−0.015)	−0.076 (−0.008)	−0.218 (−0.023)	−0.032 (−0.034)	−0.004 (−0.004)	−0.036 (−0.038)	−46.752 (−0.046)	−23.215 (−0.023)	−69.967 * (−0.070)
Age	−0.001 (−0.026)	-	−0.001 (−0.026)	−0.015 (−0.028)	−0.003 (−0.005)	−0.017 (−0.033)	−0.015 (−0.035)	−0.003 (−0.006)	−0.018 (−0.042)	−0.004 ** (−0.085)	0.000 (−0.005)	−0.004 ** (−0.089)	1.081 (0.024)	−2.400 ** (−0.053)	−1.319 (−0.029)
Frequency of using assistive device *(No need = 0)*	−0.003 (−0.010)	-	0.003 (0.010)	0.113 (0.026)	0.008 (0.002)	0.121 (0.027)	0.186 (0.051)	0.008 (0.002)	0.194 (0.054)	0.029 ** (0.082)	0.002 (0.005)	0.031 ** (0.087)	5.490 (0.014)	20.150 ** (0.052)	25.640 * (0.067)
Family income level in local society *(Low = 0)*	−0.042 * (−0.079)	-	−0.042 * (−0.079)	0.055 (0.008)	−0.102 * (−0.015)	−0.047 (−0.007)	−0.132 (−0.023)	−0.108 * (−0.019)	−0.240 (−0.042)	−0.019 (−0.035)	−0.003 (−0.005)	−0.022 (−0.040)	−12.240 (−0.020)	−15.027 (−0.025)	−27.267 (−0.045)
Years of education	0.006 (0.051)	-	0.006 (0.051)	−0.025 (−0.018)	0.013 (0.010)	−0.011 (−0.008)	−0.080 (−0.070)	0.014 (0.012)	−0.066 (−0.058)	0.003 (0.028)	0.000 (−0.004)	0.003 (0.024)	10.976 ** (0.090)	1.281 (0.011)	12.257 ** (0.100)
Residential region *(East China = 0)*	0.010 (0.024)		0.010 (0.024)	0.234 (0.041)	0.025 (0.005)	0.259 (0.046)	0.218 (0.047)	0.027 (0.006)	0.245 (0.053)	−0.003 (−0.007)	0.003 (0.006)	−0.001 (−0.002)	106.771 ** (0.216)	0.985 (0.022)	107.756 ** (0.218)

Note: The values in parentheses are standardized effects, while those outside are unstandardized effects. Statistically significant values are shown in cells with background color. * *p* < 0.05; ** *p* < 0.01.

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
