# Peer review of "Determining How Far an Adult Rare Disease Patient Needs to Travel for a Definitive Diagnosis: A Cross-Sectional Examination of the 2018 National Rare Disease Survey in China"

_ijerph, 2020, doi:10.3390/ijerph17051757_

Round 1
Reviewer 1 Report
The study reports the results of a well-designed analysis of factors influencing the accessibility to the definite diagnosis in case of rare diseases in China. Although in general, the work is compelling, there are several issues which require explanation and/or additional apart from addressing them in the limitations.
The paper lack details on how the referrals work in the health care system in China. Specifically, it would be important to understand what the rules are for the patients to be admitted to the 3-A hospital. Do patients need referrals from a lower reference centre, or they can go directly to 3-A hospital on their own? If they need a referral, what type of facility/professional is authorized to yield it?
The way subjects were recruited to the survey requires further explanation. If the questionnaire was distributed via specific patients groups, they should be enlisted in the supplementary file to the manuscript. Were questionnaires filled on paper, online or there were telephone-based?
Why finally, only 1010 individuals from Mainland China from initial 1135 respondents were included in the study? For a reader from other regions of the world, the concept of Mainland China maybe not clear? Does it mean that people living on islands were not included?
How is it possible that in ab. 1000 subjects sample, there 68 with PJS which is assessed by Authors themselves as “an extremely rare”? Apparent low representativeness of the sample for 20 mln. The population of persons with rare diseases may be an essential bias which should be discussed.
Furthermore, as the structure of the study sample seems to be not representative for the prevalence of specific diseases, one would appreciate the information not only about the prevalence but also about the frequency in the study sample in the Table A1. Apart from this, do the data on prevalence form orpha.net are specific for China or the general world population. If the latter, one would appreciate the data for China, if available.
It should be explained what factors from Andersen’s model were not included in the model developed in the study, and why?
Indicators of accessibility to definitive diagnosis – the explanation should be provided how three studies performed in Europe or North America can be extrapolated to the situation in China. Furthermore, please explain if there is no indicator specific to the health care situation in China?
Finally, isn’t there a context of competition between academic and alternative/complementary/traditional medicine or not in Mainland China? I mean if the delay in obtaining the final diagnosis is not increased by using informal health care resources?
Wasn’t the place of residence considered as a factor in the model? Were only subjects from urban areas surveyed and none from a rural area? Apparently not? Aren’t people from rural areas deprived of easy access to the health care system? Maybe it would also be important to adjust the model for the distance from the place of residence to the nearest larger city?
It is not clear what the rationale for the use of a variable “degree of dependence on assistive device” with responses based on the frequency scale.
How were the responses about family income level obtained and if, self-assessed by the respondents what the “average” level was supposed to mean?
The software tools used for generation of the map included as figure 2 and travel trajectories in figure 4 were not acknowledged.
Figure 2 - is it about the number of 3-A hospitals in “each city” or “province”?
Author Response
Dear Reviewer,
Thanks very much for taking the time to review this manuscript. We really appreciate all your comments and suggestions. Please find our itemized responses below and the revisions/corrections in the re-submitted files.
Please see the attachment.

Reviewer 2 Report
Dear Editor thank you for asking me to review this paper
I am very interested in this topic
It is a good work but it needs some revisions
First of all about English language I give you some examples:
line 39 it is missed quality of life
line 53 maybe it is missed to obtain
lines 58 and 62 references are missed
line 88, 89, 90 it is not lear what authors would say
and so on
Second It is not mentioned the ethical approval of this reseach
and then
in the abstract the background it is not clear stated there is only the aim of the study
In the matherial it si citated the income level but no information about it it is stated
Author Response

(The authors gave the same response as above.)
